# The 3D Printing Behavior of Photocurable Ceramic/Polymer Composite Slurries Prepared with Different Particle Sizes

**DOI:** 10.3390/nano12152631

**Published:** 2022-07-30

**Authors:** Kukhyeon Ryu, Jinho Kim, Junghoon Choi, Ungsoo Kim

**Affiliations:** Korea Institute of Ceramic Engineering & Technology, Icheon 17303, Korea; cj2303@haver.com (K.R.); jino.kim@kicet.re.kr (J.K.); eunu@kicet.re.kr (J.C.)

**Keywords:** ceramic 3D printing, particle size, slurry properties, photoinitiator concentration, photocuring

## Abstract

Ceramic polymer composite slurries were prepared using nano- and micro-sized Al_2_O_3_ in order to analyze rheological properties, sedimentation, and curing behavior. Slurries with different Al_2_O_3_ particle sizes were prepared with varying concentrations of photoinitiator, and subjected to different exposure times to prepare a printing object. All slurries exhibit shear-thinning behavior, and the viscosity increases with decreasing Al_2_O_3_ particle size. The 100 nm Al_2_O_3_ slurry is confirmed to be more sol-like, while the 500 nm and 2 μm Al_2_O_3_ slurries have a gel-like structure. As the Al_2_O_3_ particle size increases, a thick sedimentation layer forms due to rapid settling, but as the distance between particles increases, the UV light scattering reduces, and the curing rate increases. The exposure time range viable for printing, and the dimension conformity of the printed specimen with the design file, is improved by increasing the Al_2_O_3_ particle size. In the case of 500 nm and 2 μm Al_2_O_3_ slurries, the maximum heat flow, curing enthalpy, and conversion rate are high with respect to photoinitiator concentration, in the order of 1.0 > 0.1 > 3.0 wt.%. When the photoinitiator concentration exceeds 1 wt.%, it appears to affect the reactivity of the slurry.

## 1. Introduction

Digital light processing (DLP) 3D printing is a process that polymerizes a surface through a chemical reaction, by irradiating UV light on photocurable resins. The core of DLP 3D printing lies in the development of optimal resins, along with the processing equipment (printer). A photocurable ceramic slurry is prepared by mixing a monomer, ceramic powder, and a photoinitiator (PI). The properties of the reactive monomer and ceramic powder determine the physical and curing properties of the ceramic polymer composite slurry [1,2,3]. In addition, it affects the physical properties and characteristics of the printing object after curing. Therefore, the selection of the monomer and ceramic powder should be based on the field of application. After that, the mixing ratio of the monomer and ceramic powder, the type and concentration of PI, and additional additive composition are determined.

Many studies report and generalize the effect of powder properties on a slurry, green body, and sintered body in conventional ceramic processing [4,5,6,7], but in the case of the 3D printing photocuring process, which received attention recently, studies on the composition of the ceramic polymer composite and process conditions are being conducted [8,9,10]. In general, the effect of ceramic powder on the composite slurry and curing properties increases at a higher solid content. The solid content is the most important factor in determining the viscosity of the photocurable slurry, whereas an increase in powder content decreases transmittance and curing efficiency [11,12,13]. Particle size also acts as a major factor. As the particle size increases, the viscosity decreases; however, over time, sedimentation occurs, and, hence, sub-micron-sized powder is used. Meanwhile, if the particle size is too small, scattering increases, resulting in the adverse effect of lowering the curing degree.

Although the photocurable ceramic/polymer composite is widely applied to industrial materials, the effect of the filler size on the photocuring process and mechanical and optical properties after curing is not clearly defined. In this study, ceramic polymer composite slurries were prepared using nano- and micro- sized Al_2_O_3_, in order to analyze rheological properties, sedimentation behavior, and curing behavior. In addition, the slurries with different PI concentrations for each Al_2_O_3_ particle size were prepared, and subjected to different exposure times to prepare a printing object. From the above results, the correlation between slurry composition (Al_2_O_3_ and PI), properties (viscosity, precipitation behavior, and photocurability), and printing results was determined. Accordingly, the range of photocurable slurry composition and processing conditions to be considered according to the particle size of the powder were discussed for successful 3D printing.

## 2. Materials & Methods 

### 2.1. Materials

In this experiment, three alumina powders with different average particle sizes were used: 100 nm (26N-0811UPA, Inframat Advanced Materials, Manchester, CT, USA), 500 nm (AES-11H, Sumitomo, Tokyo, Japan), and 2 μm (AES-23, Sumitomo). Reported properties of alumina powders are summarized in Table 1. Methyl alcohol (CH_3_OH, Extra pure, DEAJUNG, Siheung-si, Korea) was used as a solvent. The silane coupling agent was 3-trimethoxy-silylpropane-1-thiol (MPTMS), with an average molecular weight of Mw = 196.34 g/mol (Product No. KBM 503, Shin-Etsu Chemicals Co., Ltd., Tokyo, Japan). The monomer was trimethylolpropane triacrylate (TMPTA), with an average molecular weight of Mw = 296.32 g/mol. As a PI, phenylbis (2,4,6-trimethyl benzoyl) phosphine oxide (Irgacure 819) was purchased from Sigma Aldrich. 

### 2.2. Al_2_O_3_ Acrylate Composite Slurry Preparation

Methyl alcohol and Al_2_O_3_ were mixed in a ratio of 6:4, and ball-milled for 24 h. A total of 20 wt.% of MPTMS with respect to Al_2_O_3_ was added, and linked to the Al_2_O_3_ surface by hydrolysis and condensation reaction through heating at 50 °C for 24 h, using a heating mantle. TMPTA was added to the MPTMS-modified Al_2_O_3_ slurry, and, subsequently, a chain polymerization reaction between MPTMS and TMPTA was induced at room temperature for 12 h. The solvent was evaporated at 50 °C using a vacuum evaporator (N-1200A, EYELA, Tokyo, Japan) to obtain a solvent-free slurry. Then, the PI was mixed for 24 h to prepare a solvent-free Al_2_O_3_ acrylate composite slurry. The final solid content of the slurry was 60 wt.% (29.5 vol.%).

### 2.3. Characterizations

The rheological behavior of the slurry was analyzed using a rheometer (HAAKE MARS III, Thermo Fisher Scientific Inc., Braunschweig, Germany). The experiment was conducted at a constant temperature of 25 °C by a cup and bob (Coaxial Cylinders)-type measuring system. Viscosity was measured in a shear rate range of ~10^−4^–10^3^ s^−1^ in a steady-state mode. Oscillation sweep (frequency and amplitude oscillation sweep) was used to evaluate the deformation behavior and viscoelastic properties of the slurry. The frequency oscillation sweep was performed at an angular frequency of 0.1–100 rad/s within the linear viscoelastic range (LVR) of the slurry, determined through the amplitude oscillation sweep measurement.

For dispersion stability analysis, a sedimentation test was performed on the slurry using a Turbiscan LAB stability analyzer (Formulaction SA, Toulouse, France). The measurement used a cylindrical glass cell and a single wavelength near-infrared laser at 880 nm as a light source. Among the measurement methods of transmission (%) and backscattering (%), the backscattering (%) profile by the scattered light was used. Backscattering value (ΔBS) depicts destabilization phenomena. Destabilization phenomena include particle migration (creaming or sedimentation) and particle size variation (flocculation or coalescence).

The curing behavior of the slurries was analyzed using a photo–differential scanning calorimeter (DSC 204 F1 phoenix, Netzsch, Selb, Germany). The specimens were placed in an aluminum crucible with a pierced lid and measured at the same temperature (25 °C), light source (405 nm), light intensity (5.4 W/cm^2^), and exposure time (20 s). The curing conversion rate was calculated based on the curing enthalpy (ΔH) from photo–DSC [14].

The printing behavior of the solvent-free Al_2_O_3_ acrylate composite slurry was determined using a DLP 3D printer (IM96, Carima, Korea). The light intensity of UV LED was 7.8 mW/cm^2^, and the wavelength was 405 nm. A disk-shaped specimen (diameter = 10 mm, thickness = 2 mm) was printed, and the dimensions of the object were measured using a vernier caliper to confirm a deviation from the designed values. The initial exposure time during printing was 20 s, while the basic exposure time for printing was changed to the following durations: 1.2, 2.4, 4.8, 9.6, and 19.2 s. Here, the initial exposure time refers to the time until the slurry is cured and adhered to a build platform, which refers to the 10-step process in the entire printing process (204 steps), while the basic exposure time refers to the exposure time for all subsequent stages after the initial exposure time.

## 3. Results and Discussion

### 3.1. Rheological Behavior 

Figure 1 shows the rheological behavior of the Al_2_O_3_ acrylate composite slurry according to particle size. The viscosity of all slurries decreases with increasing shear rate, and it increases with decreasing particle size at the same shear rate. When the particle size decreases, the surface area of the Al_2_O_3_ particles increases relatively at the same solid content, so interparticle interaction increases significantly and the viscosity increases [15]. All slurries exhibit Newtonian behavior at low shear rates, and they then show shear-thinning behavior, characterized by a decreasing viscosity at higher shear rates. This is because the particles in the slurry establish a directional arrangement at higher shear rates, decreasing interparticle interaction and viscosity [15,16,17]. This rheological property is desirable for 3D-printing slurries, owing to the low viscosity even under the stress generated during printing.

To analyze the deformation behavior and viscoelastic properties of the slurry, oscillatory shear measurement was performed (Figure 2). The 100 nm Al_2_O_3_ slurry shows a liquid-like behavior (G″ > G′) at a low frequency, and then a solid-like behavior (G″ < G′) from 40 Hz after intersection. This phenomenon means that the slurry has a more sol-like state. The 500 nm and 2 μm Al_2_O_3_ slurries exhibit solid-like behavior within the measured frequency range. In addition, both G′ and G″ tend to increase with frequency, indicating a typical gel-like slurry. 

Figure 3 shows the change in loss tangent (tan δ) with respect to frequency. The loss tangent reflects the ratio of viscosity to elasticity (energy loss to energy stored) and the damping (attenuation) of the Al_2_O_3_ slurry [16,17,18,19,20]. As the particle size increases, the loss tangent tends to decrease. The 100 nm Al_2_O_3_ slurry shows a significant decrease in loss tangent at higher frequencies. This is because small particles in the slurry are weakly agglomerated, and behave similar to irregular single particles that easily roll or slip during shear deformation, resulting in a high viscosity in comparison with their elasticity [18,19].

### 3.2. Sedimentation Behavior

The sedimentation behavior of Al_2_O_3_ acrylate composite slurry was observed for 24 h to determine the change in dispersion stability (Figure 4). Unlike other ceramic processing technologies, the 3D printing process repeatedly stacks layers with thicknesses in the range of several tens of microns, and requires a relatively long process time. Therefore, maintaining the uniformity of the slurry during forming can act as a factor that directly affects the quality of the object.

The glass cell with specimen was divided into a top, middle, and bottom layer according to height, and the dispersion stability of the slurry was confirmed based on the change in the backscattering profile of each layer. Rayleigh diffusion is observed in the upper layer of the 100 nm Al_2_O_3_ slurry, confirming a clarification zone in which backscattering sharply increases. Rayleigh diffusion occurs when the particle size is smaller than the wavelength of the light source (<0.5 μm); as the distance between small particles becomes narrow, the scattering probability between light and particles increases, resulting in an increase in backscattering (%) [21,22]. A slight increase in backscattering over time in the middle and bottom layers indicates relatively weak aggregation and sedimentation.

In the case of 500 nm and 2 μm Al_2_O_3_ slurries, a clarification with reduced backscattering is observed in the top layer. When the particle size is larger than the wavelength of light used as a light source (>0.5 μm), Mie diffusion occurs, and as the particle size increases, the average interparticle distance also increases, resulting in fewer scattering events between light and particles [21,22,23]. As a result, the backscattering (%) value of the 2 μm Al_2_O_3_ acrylate composite slurry with the largest particle size is further reduced.

The Turbiscan Stability Index (TSI) and peak thickness of the slurry were measured, and are shown in Figure 5. The TSI is an accumulation of the variations in the dispersion stability over time with the integrated backscattering profile area. It always appears as an absolute value and increases to a (+) value. That is, as the TSI value increases, the dispersion stability worsens. The TSI value increases sharply with increasing particle size, indicating that sedimentation occurs rapidly at larger particle sizes. The 100 nm Al_2_O_3_ slurry was observed for 24 h and has almost no sedimentation. 

Peak thickness is expressed by quantifying the phase thickness separated by particle migration such as sedimentation. In other words, the larger the peak thickness, the thicker the sedimentation layer, due to agglomeration. The thickness of the 100 nm Al_2_O_3_ slurry is about 0.25 mm at 0–20 h; it then becomes steady, and decreases again to 0 mm from 20 h onward. Meanwhile, the peak thicknesses of the 500 nm and 2 μm Al_2_O_3_ slurries show no change in the beginning; they then tend to increase over time. In the case of the 500 nm Al_2_O_3_ slurry, the thickness is approximately 1.0 mm after 24 h, while in the case of the 2 μm Al_2_O_3_ slurry, the thickness starts to increase rapidly from 8 h onward, and reaches a high value of approximately 2.0 mm after 24 h. Thus, it is confirmed that the sedimentation layer increases proportionally with Al_2_O_3_ particle size.

### 3.3. Curing Behavior

The curing behavior of Al_2_O_3_ acrylate composite slurry is summarized in Table 2 and Figure 6. Photo–DSC measurements were conducted in an isothermal mode using an exposure time of 20 s, the same as the basic exposure time used during printing. When the Al_2_O_3_ particle size decreases, the exothermic peak also decreases. In the 100 nm Al_2_O_3_ slurry with the smallest particle size, the dispersed particles cause UV light scattering and absorption during photopolymerization, resulting in a shorter cure depth [13]. As a result, the value of the maximum heat flow (ΔQ_max) decreases. In other words, the photon transport mean free path decreases with Al_2_O_3_ particle size, resulting in a proportional decrease in the maximum heat flow (ΔQ_max).

When comparing the curing enthalpy according to particle size, the 2 µm Al_2_O_3_ slurry shows the highest value, and, contrary to expectations, the 100 nm Al_2_O_3_ slurry shows higher values than those for the 500 nm Al_2_O_3_ slurry. These results might be due to the difference in curing time. The 100 nm Al_2_O_3_ slurry with a relatively broad heat flow takes approximately two minutes to complete curing (at the point where the heat flow becomes zero). On the other hand, the curing of the 500 nm Al_2_O_3_ slurry is achieved in less than one minute. Thus, the curing enthalpy value represented by the area in the time vs. heat flow graph is higher for the 100 nm Al_2_O_3_ slurry with a larger area.

For the 500 nm and 2 μm Al_2_O_3_ slurries, the values of maximum heat flow, curing enthalpy, and conversion % are high according to PI concentration, in the order of 1.0 > 0.1 > 3.0 wt.%. The PI irradiated with UV light absorbs photons and is excited to a higher energy state, resulting in high chemical reactivity (sensitivity) with the monomer [14]. As the PI concentration exceeds the appropriate amount (~1 wt.%), the damping coefficient increases based on the Beer–Lambert law and, consequently, affects the reactivity of the slurry. Therefore, it is determined that the values of maximum heat flow, curing enthalpy, and conversion % decrease when the PI concentration exceeds 1 wt.%.

Using the measured curing enthalpy (ΔH), the curing conversion rate according to particle size and PI concentration was calculated, and is shown in Table 2 and in the right graph of Figure 6. As the particle size increases, the difference in conversion rate according to PI concentration is found to be significant. This means that as the particle size increases for the same solid content, the distance between particles becomes wider, and the concentration of PIs between them relatively increases, leading to an increase in photopolymerization reactivity. On the other hand, when the particle size in the slurry is small, the reactivity decreases, owing to an increase in diffraction, scattering, and absorption of the light. Thus, the difference in photocuring conversion rate according to particle size and PI concentration is small [14,20,24].

### 3.4. Printing Behavior

Table 3 and Figure 7 summarize the effects of Al_2_O_3_ particle size and PI concentration on the printing results. As the Al_2_O_3_ particle size increases, the range of exposure time viable for printing, and the dimension conformity of the printed specimen with the design file, improves. The optimum printing results are achieved at a PI concentration of 1 wt.%. However, the dimensions of the printed specimen are confirmed to significantly deviate from the design file as exposure time increases at the same PI concentration.

In the case of the 100 nm Al_2_O_3_ slurry, the printing fails, as the printed object detaches from the build platform at the initial stage under all exposure time conditions. It can be concluded that the scattering and absorption of UV light by Al_2_O_3_ particles occurs rigorously during printing, decreasing the curing rate and affecting the printing behavior. As the particle size increases, the range of exposure time viable for printing the Al_2_O_3_ acrylate composite slurry and the conformity improves. After printing, the 500 nm Al_2_O_3_ slurry has the highest degree of conformity (105.2, 102%) at the exposure time of 2.4 s, while the 2 μm Al_2_O_3_ slurry shows the best conformity (101.7, 101.2%) at 1.2 s.

The range of exposure time viable for printing is the widest at a PI concentration of 1.0 wt.%, yielding the highest maximum heat flow value and curing enthalpy, followed by 0.1 wt.% PI concentration. The slurries containing 3.0 wt.% PI concentration, with a relatively low maximum heat flow and curing enthalpy, mostly fail to print Al_2_O_3_ acrylate composites. From the analysis of the relationship between the curing properties of the slurry and the printing result, the maximum heat flow exhibits a higher probability as an index for predicting the success of printing in comparison with the curing enthalpy and conversion % of the slurry.

The conformity of the printed specimen as a function of exposure time is lower for the 1.0 wt.% slurry compared to that of the slurry with a 0.1 wt.% PI concentration. Photocurable slurries with a high PI concentration undergo very intense photo-polymerization when exposed to UV for a long time. The curing range of the ceramic slurry can be explained by the Gaussian beam model, where the curing depth increases at a higher curing rate, consequently increasing the curing width (WGauss) [25]. Therefore, when a slurry with a high PI concentration is exposed to UV for a long time, the precision of the printed object deteriorates.

To define the correlation between curing and printing behavior of slurries with different particle sizes and PI concentrations, the X-axis in Figure 8 is set as the PI concentration, the Y-axis as the Al_2_O_3_ particle size, and the Z-axis as the maximum heat flow (ΔQmax), to indicate the viable and non-viable area for printing. When the Al_2_O_3_ particle size increases, or the PI concentration decreases, the ΔQmax value tends to increase, indicating the viable area for printing. The highest maximum heat flow (6.72 mw/mg) is achieved at the Al_2_O_3_ particle size of 2 μm and PI concentration of 1.0 wt.%, and in this vicinity, excellent printing properties are exhibited.

## 4. Conclusions

In this study, the effect of Al_2_O_3_ particle size (100 nm, 500 nm, and 2 μm) on the properties of ceramic polymer composite slurries and 3D printing results was determined. Steady-state mode analysis confirms that all slurries exhibit a shear-thinning behavior. The 100 nm Al_2_O_3_ slurry appears to be more sol-like, while the 500 nm and 2 μm Al_2_O_3_ slurries have a gel-like structure. With increasing Al_2_O_3_ particle size, the thickness of the sedimentation layer increases, owing to rapid sedimentation, but the effect on printing within 8 h is concluded to be negligible. The distance between particles increases with Al_2_O_3_ particle size, resulting in a longer photon transport mean free path and a photocurability improvement. Consequently, the range of exposure time viable for printing, and the dimension conformity of the printed specimen with the design file, improves when the Al_2_O_3_ particle size increases. For the 500 nm and 2 μm Al_2_O_3_ slurries, the values of maximum heat flow, curing enthalpy, and conversion rate are high according to the photoinitiator concentration, in the order of 1.0 > 0.1 > 3.0 wt.%. When the concentration of the photoinitiator exceeds the appropriate amount of 1 wt.%, the damping coefficient increases, according to the Beer–Lambert law, and affects the reactivity of the slurry.

## Figures and Tables

**Figure 1 nanomaterials-12-02631-f001:**
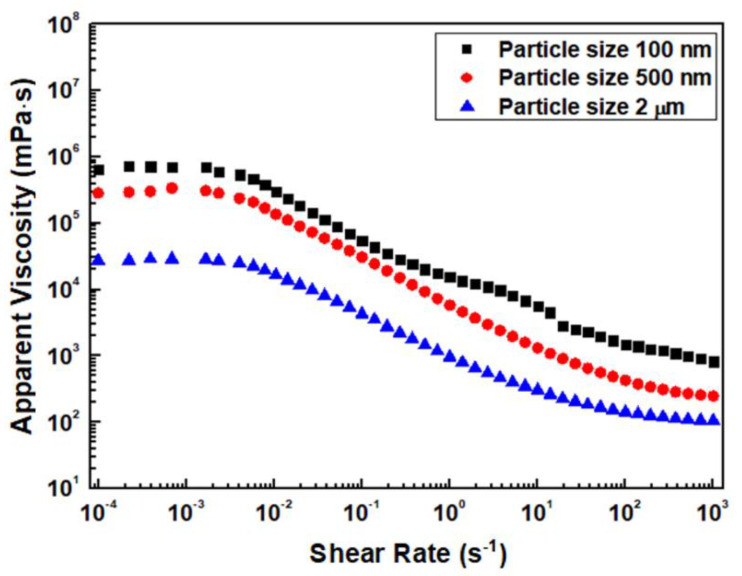
Rheological behavior of Al_2_O_3_ slurries prepared with different particle sizes.

**Figure 2 nanomaterials-12-02631-f002:**
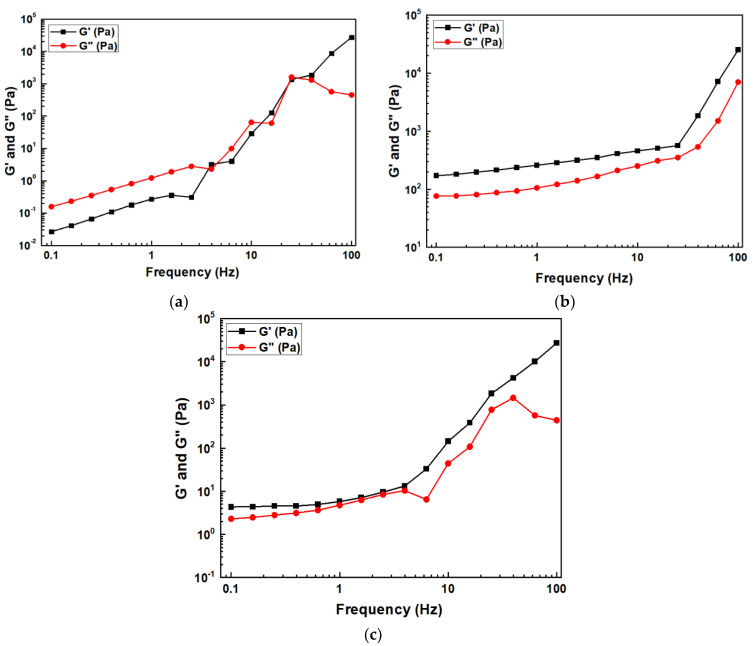
Particle size effect (**a**) 100 nm, (**b**) 500 nm, and (**c**) 2 μm on the frequency sweep of Al_2_O_3_ slurries.

**Figure 3 nanomaterials-12-02631-f003:**
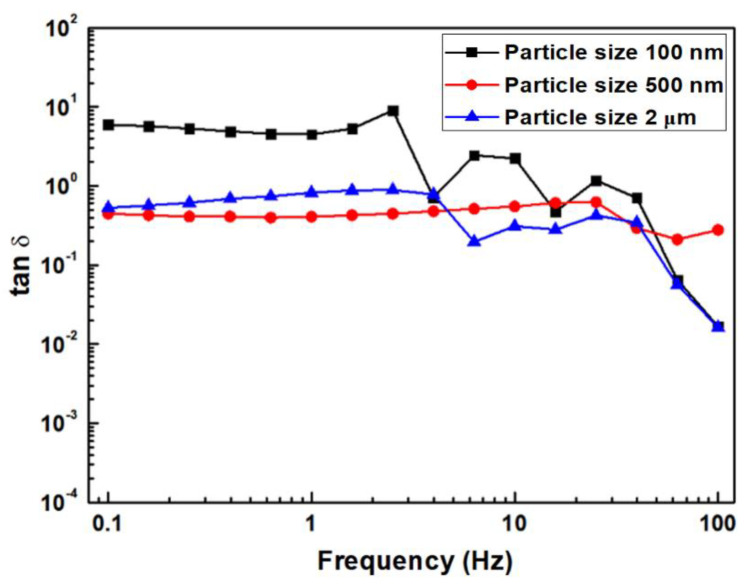
tan δ values of Al_2_O_3_ slurries as a function of particle size.

**Figure 4 nanomaterials-12-02631-f004:**
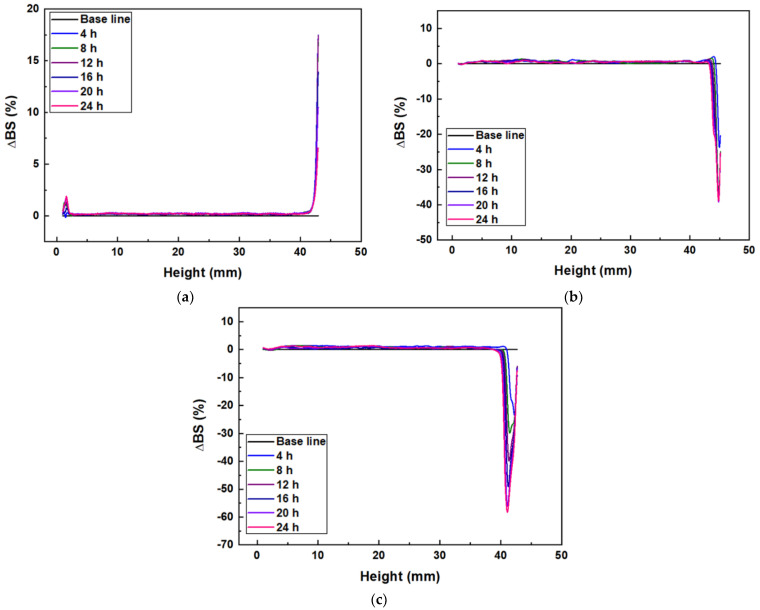
Backscattering profiles of Al_2_O_3_ slurries with different alumina particle sizes: (**a**) 100 nm, (**b**) 500 nm, and (**c**) 2 μm.

**Figure 5 nanomaterials-12-02631-f005:**
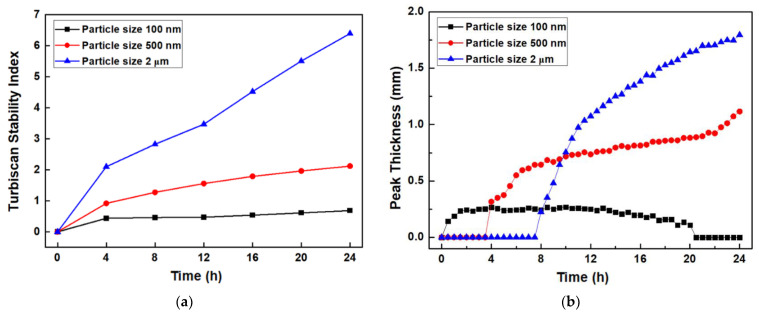
(**a**) TSI curves, (**b**) peak thickness of Al_2_O_3_ slurries with different particle sizes.

**Figure 6 nanomaterials-12-02631-f006:**
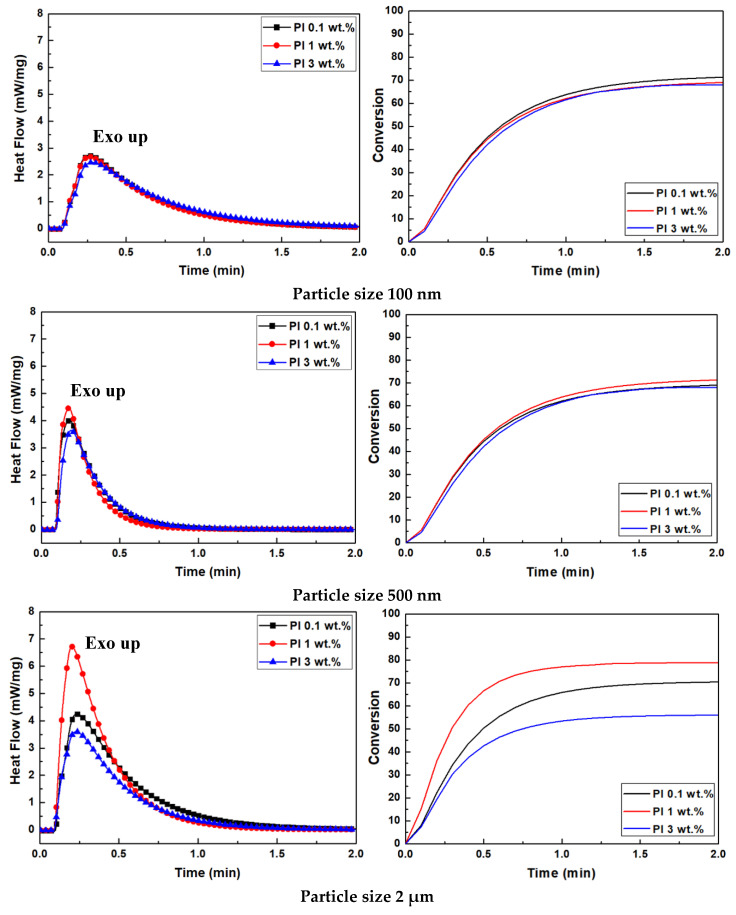
Heat flow and conversion of photocurable Al_2_O_3_ slurries with different particle sizes obtained by photo–DSC. Photocurable slurries were prepared with acrylate monomers with concentrations of 0.1, 1, and 3 wt.% Irgacure 189 photoinitiator.

**Figure 7 nanomaterials-12-02631-f007:**
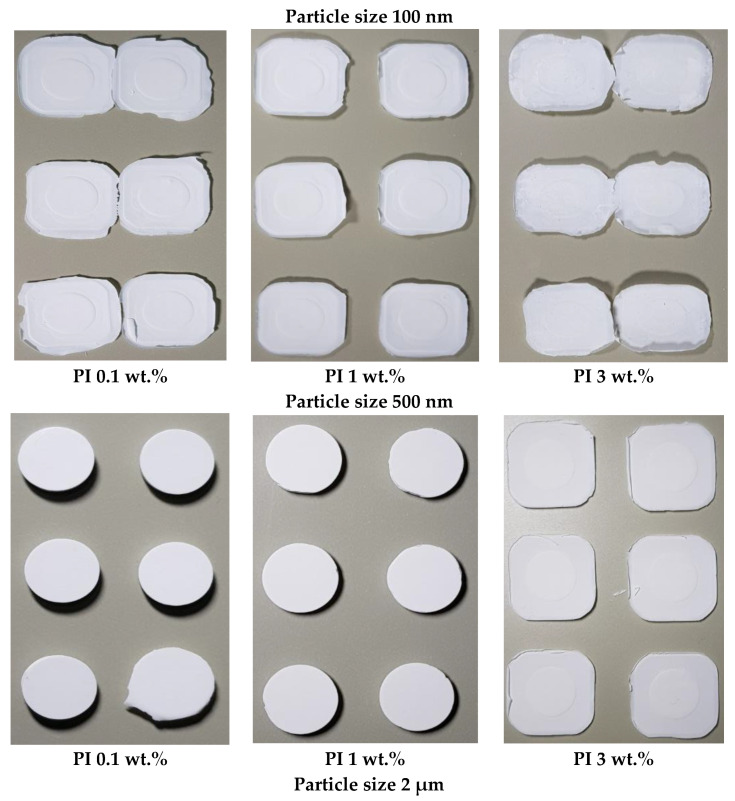
Printed objects using the photocurable Al_2_O_3_ slurries with different particle sizes at 4.8 s of basic exposure time.

**Figure 8 nanomaterials-12-02631-f008:**
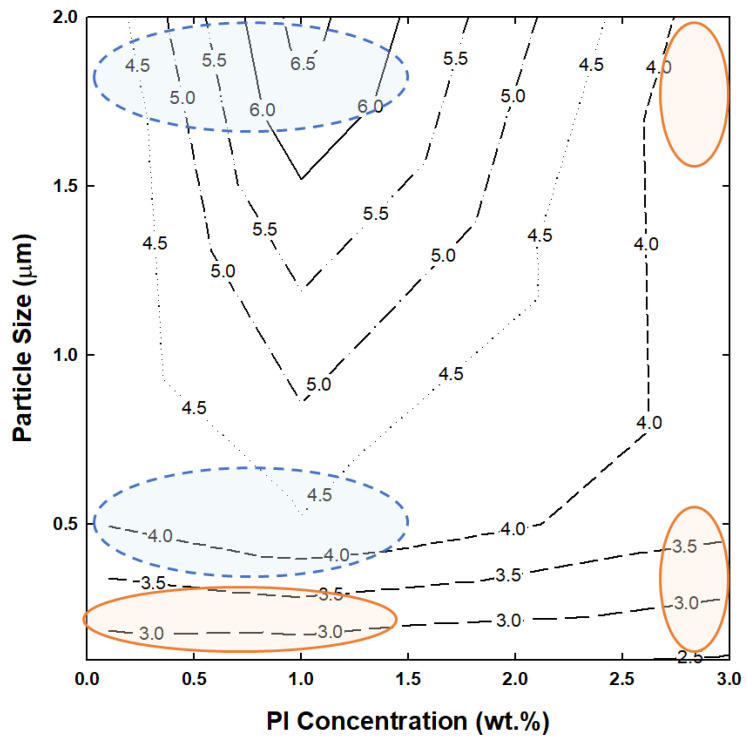
Relationship between particle size, photoinitiator concentration, maximum heat flow, and printability. The blue circles indicate the areas where printing is successful, while the red circles indicate areas where printing fails.

**Table 1 nanomaterials-12-02631-t001:** Reported properties of Al_2_O_3_ powders used in this study.

	26N-0811UPA	AES-11H	AES-23
D_50_	100 nm	0.54 μm	2.2 μm
Specific surface area (m^2^/g)	13–15	5.60	4.22
α crystal size (μm)	NA	0.3	0.3–4
Density (g/cm^3^)	3.97	3.97	3.97
Al_2_O_3_ purity (%)	99.99	99.9	99.9

**Table 2 nanomaterials-12-02631-t002:** Cure parameters for photo-polymerization reactions at various particle sizes and photoinitiator concentrations (exposure time = 20 s).

Particle Size	Photoinitiator Concentration (wt.%)	Maximum Heat Flow (mW/mg)	Curing Enthalpy (J/g)	Conversion (%)
100 nm	0.1	2.72	90.42	71
	1	2.68	89.60	69
	3	2.46	87.60	67
500 nm	0.1	4.02	61.44	69
	1	4.46	66.21	71
	3	3.64	59.99	67
2 μm	0.1	4.25	122.6	70
	1	6.72	137.3	79
	3	3.60	95.56	56

**Table 3 nanomaterials-12-02631-t003:** Deviation of printing objects from the original design file according to slurry composition (particle size and PI) and exposure time.

Particle Size	Photoinitiator Concentration (wt.%)	Concordance to Design (%)				
		Initial Exposure Time (s)				
		1.2	2.4	4.8	9.6	19.2
500 nm	0.1	-	-	103.6	104.7	109.2
	1	-	105.2	110.6	130	117.6
	3	-	102	-	-	-
2 μm	0.1	X	105.8	99.2	107.5	113.6
	1	101.7	105.8	110.5	118.4	111.4
	3	101.2	-	-	-	-

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
