# Peer review of "The 3D Printing Behavior of Photocurable Ceramic/Polymer Composite Slurries Prepared with Different Particle Sizes"

_nanomaterials, 2022, doi:10.3390/nano12152631_

Round 1
Reviewer 1 Report
Dear Authors,
The article details the rheology, sedimentation, curing behavior and printability of slurries with different Al2O3 particle sizes. Such slurries are potentially suitable for 3D printing.
This paper is of high scientific quality and can be accepted after minor revisions. The following comments, however, should be addressed before publication.
1. Please provide the key characteristics of each aluminum oxide powders, such as BET surface, crystallite size, tapped density etc. The powders used have clearly different specific surface areas, but no values are given in the text.
2. Why is the same concentration of dispersant (20 wt.% of MPTMS) used for all powders? Is this value too high? Typically, the amount of dispersant depends on the specific surface area of the powder and value 1-5 wt.%.
3. Dispersant MPTMS content a lot of Silicon and Sulfur atoms. What is the chemical composition of the final ceramic after high temperature firing?
4. The final solid content of the slurry was 60 wt.%. What is the volume fraction? About 30 vol.%? Please add calculated value to the text.
5. In the text, you operate everywhere with time (1-20 seconds), but nowhere do you indicate the light power of your printer. Please add to the text the light intensity (in mW/cm2) for DLP 3D printer IM96. This is necessary to understand the absolute energy values.
6. What is the curing thickness (Cure depth versus time or energy dose) of the all suspensions?
Author Response
Dear Reviewer #1
Thank you for reviewing our manuscript and for providing us with your comments. Our point-by-point responses to your comments are set out below. We have made every effort to fully address all of your concerns and hope the manuscript is now acceptable for publication in the Nanomaterials. If you require any further information, please do not hesitate to contact me.
1. Please provide the key characteristics of each aluminum oxide powders, such as BET surface, crystallite size, tapped density etc. The powders used have clearly different specific surface areas, but no values are given in the text.
(Response)
Table 1 is added with the reported properties of three different Al2O3 powders.
|
26N-0811UPA |
AES11-H |
AES-23 |
D50 |
100 nm |
0.54 μm |
2.2 μm |
Specific Surface Area (m2/g) |
13 - 15 |
5.60 |
4.22 |
α Crystal Size (μm) |
NA |
0.3 |
0.3 - 4 |
Density (g/cm3) |
3.97 |
3.97 |
3.97 |
Al2O3 Purity (%) |
99.99 |
99.9 |
99.9 |
2. Why is the same concentration of dispersant (20 wt.% of MPTMS) used for all powders? Is this value too high? Typically, the amount of dispersant depends on the specific surface area of the powder and value 1-5 wt.%.
(Response)
The concentration of 20 wt.% was determined from our unpublished preliminary study. In rheological study rapid decrease in viscosity was shown until the addition of 10 wt.% MPTMS, and then slow decrease in viscosity was continued by 20 wt.%. It was found that adsorption amount of MPTMS on Al2O3 particles was increased up to 20 wt.% using TG-DTA analysis.
3. Dispersant MPTMS content a lot of Silicon and Sulfur atoms. What is the chemical composition of the final ceramic after high temperature firing?
(Response)
We have not yet analyzed the chemical composition of fired specimens. However, we assume that after thermal decomposition sulfur can be released in the form of SO2 gas, and silicon can react with alumina to form mullite phase in the grain boundary.
4. The final solid content of the slurry was 60 wt.%. What is the volume fraction? About 30 vol.%? Please add calculated value to the text.
(Response)
60 wt.% corresponds to 29.5 vol.%. This is added in the text.
5. In the text, you operate everywhere with time (1-20 seconds), but nowhere do you indicate the light power of your printer. Please add to the text the light intensity (in mW/cm2) for DLP 3D printer IM96. This is necessary to understand the absolute energy values.
(Response)
The following is added in the text. The light intensity of UV LED was 7.8 mW/cm2 and the wavelength was 405 nm.
6. What is the curing thickness (Cure depth versus time or energy dose) of the all suspensions?
(Response)
Curing thickness was not measured using the slurries. Instead, it can be estimated from the thickness of the printed object and the number of repeated printing steps. Depending on the slurry composition and exposure time the curing thickness was varied in the range of 10.46 and 17.53 μm.

Reviewer 2 Report
The manuscript entitled “3D Printing Behavior of Photocurable Ceramic/Polymer Composite Slurries Prepared with Different Particle Sizes” has been reviewed. The results are helpful. However, the manuscript should be well improved before acceptance. Detailed comments are as follows:
1. The manuscript was not well prepared. The quality of figures and writing style should be better.
2. The objective needs to be improved. Why studied the ceramic polymer slurries with different Al2O3 particle sizes?
3. Some abbreviations, such as PI and DSC, in the main text must be defined at their first mention there.
4. There are many typo errors in the manuscript. Units should be separated from the numerical value by a space, especially in Figures. G’ and G’’ should be G' and G''. Axis names should be separated from the left-half brackets by a space in some Figures. Please double-check your manuscript.
5. In all subtitles, “of Al2O3 acrylate composite slurry” should be removed.
6. What is TSI? This should be defined in the main text.
7. “heat flow rate” in Table 1 and the main text should be “heat flow”.
8. Wt.% or wt% should be unified as wt.%.
9. Please unify the capitalization of first letters in Tables and Figures.
10. In Fig. 2, in Pa should be (Pa). f Hz should be Frequency (Hz).
11. In Fig. 3, tan (delta) should be tan delta. f Hz should be Frequency (Hz). In the figure caption, Tan delta should be tan delta, since "tan" is a trigonometric operator and should therefore not be capitalized.
12. In Fig. 4, Base line should be Baseline.
13. In Fig. 5a, the unit of TSI should be added.
14. P.I should be PI in Figs. 6 and 7.
15. In Fig. 6, % should be wt.%. conversion % should be conversion.
16. In Fig. 7, wt% should be wt.%.
17. In Fig. 8, P.I. should be PI.
Author Response
Dear Reviewer #2
Thank you for reviewing our manuscript and for providing us with your comments. Our point-by-point responses to your comments are set out below. We have made every effort to fully address all of your concerns and hope the manuscript is now acceptable for publication in the Nanomaterials. If you require any further information, please do not hesitate to contact me.
1. The manuscript was not well prepared. The quality of figures and writing style should be better.
(Response)
Figures and writing style were modified for better readability.
2. The objective needs to be improved. Why studied the ceramic polymer slurries with different Al2O3 particle sizes?
(Response)
The paragraph including the objective is modified as follows. Although the photocurable ceramic/polymer resin composite has been widely applied to industrial materials, the effect of the filler size on photocuring process and mechanical and optical properties after curing is not clearly defined. In this study, ceramic polymer composite slurries were prepared using nano- and micro- sized Al2O3 to analyze rheological properties, sedimentation behavior, and curing behavior. In addition, the slurries with different photoinitiator concentrations for each Al2O3 particle size were prepared and subjected to different exposure times to prepare a printing object. From the above results, the correlation between slurry composition (Al2O3 and photoinitiator), properties (viscosity, precipitation behavior, and photocurability), and printing results was determined. Accordingly, the range of photocurable slurry composition and processing conditions to be considered according to the particle size of the powder were discussed for successful 3D printing.
3. Some abbreviations, such as PI and DSC, in the main text must be defined at their first mention there.
(Response)
PI is defined in page 2 and DSC is defined in page 4 at their first mention.
4. There are many typo errors in the manuscript. Units should be separated from the numerical value by a space, especially in Figures. G’ and G’’ should be G' and G''. Axis names should be separated from the left-half brackets by a space in some Figures. Please double-check your manuscript.
(Response)
Units are separated from the numerical values by a space. G’ and G’’ are corrected as G' and G''. Axis name is separated from the left-hand bracket by a space in Figure 4.
5. In all subtitles, “of Al2O3 acrylate composite slurry” should be removed.
(Response)
In all subtitles, “of Al2O3 acrylate composite slurry” was deleted.
6. What is TSI? This should be defined in the main text.
(Response)
The following is added in the text of page 6.
The TSI is an accumulation of the variations in the dispersion stability over time with the integrated backscattering profile area. It always appears as an absolute value and increases to (+) value. That is, as the TSI value increases, the dispersion stability worsens.
7. heat flow rate” in Table 1 and the main text should be “heat flow”.
(Response)
All the “heat flow rate” in a Table and the main text is corrected as “heat flow”.
8. Wt.% or wt% should be unified as wt.%.
(Response)
All the Wt.% and wt% were corrected as wt.%.
9. Please unify the capitalization of first letters in Tables and Figures.
(Response)
Corrected as recommended.
10. In Fig. 2, in Pa should be (Pa). f Hz should be Frequency (Hz).
(Response)
Figure 2 was corrected as recommended.
11. In Fig. 3, tan (delta) should be tan delta. f Hz should be Frequency (Hz). In the figure caption, Tan delta should be tan delta, since "tan" is a trigonometric operator and should therefore not be capitalized.
(Response)
Figure 3 and figure caption were corrected as recommended.
12. In Fig. 4, Base line should be Baseline.
(Response)
Base line in Figure 4 was corrected as Baseline.
13. In Fig. 5a, the unit of TSI should be added.
(Response)
TSI is unitless parameter, describing a cumulative sum of all the backscattering or transmission variation of the sample.
14. P.I should be PI in Figs. 6 and 7.
(Response)
Corrected as PIs in Figures 6 and 7.
15. In Fig. 6, % should be wt.%. conversion % should be conversion.
(Response)
Corrected as wt.% and conversion.
16. In Fig. 7, wt% should be wt.%.
(Response)
Corrected as wt.% in Figure 7.
17. In Fig. 8, P.I. should be PI.
(Response)
Corrected as PI and wt.% in Figure 8.

Round 2
Reviewer 2 Report
Most comments have been revised. The manuscript can be accepted if the following comments are considered:
1. There are still some typo errors in the manuscript. PI should be separated from words by a space. Please double-check your manuscript.
2. Pay attention to the capitalization of first letters in Tables and Figures, for e.g., size and enthalpy in Table 2, rate in Fig. 1 and flow in Fig. 6
Author Response
Thank you for reviewing our manuscript and for providing us with your comments. Our point-by-point responses to your comments are set out below. We have made every effort to fully address all of your concerns and hope the manuscript is now acceptable for publication in the Nanomaterials. If you require any further information, please do not hesitate to contact me.
Most comments have been revised. The manuscript can be accepted if the following comments are considered:
1.There are still some typo errors in the manuscript. PI should be separated from words by a space. Please double-check your manuscript.
(Response)
Space before or after the word ‘PI’ was corrected. Also, space between words was corrected.
2. Pay attention to the capitalization of first letters in Tables and Figures, for e.g., size and enthalpy in Table 2, rate in Fig. 1 and flow in Fig. 6.
(Response)
Capitalization of first letter were corrected in Table 2 and 3. Capitalization of x-axis was corrected in Figure 1. Capitalization of y-axis was corrected in Figure 6.
